# Laboratory Evaluation and Field Feasibility of Micro-Encapsulated Insecticide Effect on *Rhodnius prolixus* and *Triatoma dimidiata* Mortality in Rural Households in Boyacá, Colombia

**DOI:** 10.3390/insects13111061

**Published:** 2022-11-17

**Authors:** Lídia Gual-Gonzalez, Manuel Medina, César Valverde-Castro, Virgilio Beltrán, Rodrigo Caro, Omar Triana-Chávez, Melissa S. Nolan, Omar Cantillo-Barraza

**Affiliations:** 1Arnold School of Public Health, University of South Carolina, Columbia, SC 29208, USA; 2Unidad de Control de Enfermedades Transmitidas por Vectores, Secretaría de Salud Boyacá, Tunja 150001, Colombia; 3Grupo de Investigación en Medicina Tropical, Universidad del Magdalena, Santa Marta 470003, Colombia; 4Grupo Biología y Control Enfermedades Infecciosas, Universidad Antioquia, Medellín 050010, Colombia

**Keywords:** vector control, Chagas disease, *Rhodnius prolixus*, *Triatoma dimidiata*, micro-encapsulated insecticide

## Abstract

**Simple Summary:**

Vector control strategies need to adapt due to the increase in insecticide resistance among several vector species. Insecticide spraying requires great economic and logistical efforts due to repeated application every six months. This study evaluated the feasibility of micro-encapsulated insecticide on mortality on *Rhodnius prolixus* and *Triatoma dimidiata* and its effect on household colonization after one year application in a rural community in Colombia. Findings suggest the ability of this paint to prevent colonization and demonstrated positive population perception. Complementary use of insecticide spraying and micro-encapsulated paints suggest this could be a more efficient public health approach that could reduce costs and logistical efforts.

**Abstract:**

Chagas disease is a neglected vector-borne zoonosis caused by the parasite *Trypanosoma cruzi* that is primarily transmitted by insects of the subfamily Triatominae. Although control efforts targeting domestic infestations of *Rhodnius prolixus* have been largely successful, with several regions in Boyacá department certified free of *T. cruzi* transmission by intradomicile *R. prolixus*, novel native species are emerging, increasing the risk of disease. *Triatoma dimidiata* is the second most important species in Colombia, and conventional control methods seem to be less effective. In this study we evaluated the efficacy and usefulness of micro-encapsulated insecticide paints in laboratory conditions and its applicability in rural communities to avoid triatomine domiciliation. Laboratory conditions measured mortality at 6 months and 12 months, with an average mortality between 93–100% for *T. dimidiata* and 100% for *R. prolixus*. Evaluation of triatomine infestation in rural households was measured after one year, with an overall perception of effectiveness in reducing household domiciliation. Although triatomines were still spotted inside and around the homes, our findings demonstrate the ability of micro-encapsulated insecticide to prevent colonization inside the households when comparing infestation rates from previous years. Current control measures suggest insecticide spraying every six months, which implies great economic cost and logistical effort. Complementary triatomine control measures with insecticide spraying and micro-encapsulated insecticide paint would make public health efforts more efficient and reduce the frequency of treatment.

## 1. Introduction

Chagas disease is a neglected vector-borne zoonosis caused by the parasite *Trypanosoma cruzi* primarily transmitted by hematophagous insects of the Triatominae subfamily established mostly in the Americas. Other transmission routes are less common, but include congenital transmission, blood transfusions, organ transplants, and oral transmission. It is estimated that more than 6 million people are living with infection and 70 million people live at risk of infection [1]. In Colombia, an estimated 1 to 2 million people are infected and 3.5 million people are at risk [2,3]. There, the burden of disease has an estimated cost of US $13 million due to healthcare expenses and productivity losses [4]. It mostly affects rural and poor communities with poor healthcare access and vector control infrastructure, challenging diagnostic and treatment opportunities [5]. Integrated vector control strategies are an effective measure to prevent Chagas disease, including chemical insecticide spraying in infested households, infrastructure improvement to reduce colonization, and educational campaigns to inform vulnerable communities on vector-borne infections [6,7].

Regional campaigns promoting Indoor Residual Spraying (IRS) have been carried out for the past 30 years in Latin America to eradicate the domestic infestation of triatomine species and eliminate exotic vector species. The Southern Cone Initiative (INCOSUR) in 1991 fostered the implementation of other regional initiatives [8]: the Andean Countries Initiative (ACI) [9] and the Initiatives for Chagas Disease Control in Central America and Mexico (IPCAM) [10] created in 1997 and 1998, respectively, followed by the Amazon Initiative in 2004 [11]. Elimination efforts have been largely successful, with 17 countries subsequently certified by the Pan-American Health Organization [12] to be free of vectorial transmission between 1999 and 2014 [13,14]. However, in the decade following these public health successes, other vector species have emerged and are becoming established. For example, *Triatoma dimidiata* and *T. venosa* have been recorded in areas where *Rhodnius prolixus* was eliminated [15,16], and *T. brasiliensis, T. pseudomaculata, T. sordida* and *Panstrongylus megistus* have been noted where *Triatoma infestans* was eliminated [17,18,19]. In a few instances, some of these secondary vectors of public health importance have recolonized and reinfested households after multiple treatments [17,18,19].

In Colombia, 26 triatomine species have been identified, highlighting the great species biodiversity nationally [20]. Overall, 66 municipalities in four departments, including Boyacá, were certified as “Domestic *T. cruzi* transmission by *R. prolixus* interrupted” in 2013, 2015, and 2019. This success was awarded due to the large-scale insecticide application in prioritized areas and reinfestation surveillance [13]. Insecticide spraying every six months has shown to be effective to maintain households free of *R. prolixus;* however, *Triatoma dimidiata*, the second most important vector in the region, has the ability to reinfest households previously treated, challenging the reduction of overall domestic triatomine presence [15]. Additionally, some vector-free certified municipalities have evidence of *R. prolixus* presence after insecticide application, implying there could be failure in elimination or possible insecticide resistance that needs further public health attention [21]. Boyacá, northeast of Bogotá-the capital district, is one of the most affected departments in Colombia. Intradomicile reinfestation has caused a reemergence of Chagas disease in the department. In Boyacá, 10 triatomine species have been identified, distributed across 63 municipalities; *Triatoma dimidiata*, *Triatoma venosa, Panstrongylus geniculatus*, and *Rhodnius prolixus* are the most commonly reported [22]. Triatomines persist in peridomestic and sylvatic environments, including in regions that have been certified free of vectorial transmission. Cumulatively, these emergent factors demand strategic intervention alternatives to combat ecological components maintaining these secondary and re-emerging vectors.

The primary chemicals recommended for triatomine control are λ-cyhalothrin, cypermethrin, and deltamethrin [23]. The use of these compounds as pesticides to control other agriculturally relevant pests–such as the palm weevil–have been suggested to overexpose palm-living triatomines to those insecticides, which are also recommended for Chagas vector control [24]. Limited evidence suggests long term exposure to these compounds can promote triatomine insecticide resistance, and it is becoming a concern for vector-control strategies [24]. Further, it has been hypothesized that these insecticide-resistant triatomines migrate to urbanized areas, surviving the control strategies used in traditional indoor spraying formulates [25]. Insecticide resistance is a growing concern in the region; however, most efforts are focused on *Aedes aegypti* mitigation [26,27,28]. New evidence suggests triatomine populations are also likely developing insecticide resistance, warranting novel interventions using alternative compounds for successful transmission interruption [29,30].

Insecticidal paint use is a well-known and community accepted alternative to indoor residual spraying [31,32,33]. Despite its historical use in Latin American countries, standardized protocols and rigorous studies evaluating optimal efficacy for triatomine reduction are lacking. As a strategy to prevent triatomine domiciliation, studies evaluating insecticidal paint have shown households stay free of triatomines for longer periods as compared to regular indoor domiciliary insecticide spraying, which could potentially be a successful alternative or a complementary intervention [34,35]. To understand the utility of this alternative strategy, we performed a mixed-methods study, evaluating the effect of insecticidal paint on *Triatoma dimidiata* and *Rhodnius prolixus* under laboratory conditions, and assessing the application of insecticidal paint in different households in the Boyacá department, Colombia.

## 2. Materials and Methods

### 2.1. Study Area

The study was conducted between July 2017 and December 2019, with laboratory experiments performed at the Universidad de Antioquia in Medellín and field intervention in Socotá municipalities (Boyacá department), Colombia. The Boyacá department is classified within the biogeographical zone of Sub-Tropical humid forest. The average temperature varies between 18–24 °C with an average annual precipitation between 1000–2000 mm. It is placed within the Andean mountain ranges and differences in altitudes provide a wide range of climates. Overall, 24 typical homes from the municipalities Chusvitá, Guaquira, La Vega, and Pueblo Nuevo were selected for the intervention study. Households were selected based on suitability to implement treatment, from areas with greater insect presence (Figure 1).

### 2.2. Insects

The study laboratory setting used established *R. prolixus* and *T. dimidiata* colonies from the Laboratory of Biology and Infectious Disease Control group (BCEI by its acronym in Spanish) at the Universidad de Antioquia, Medellín. Fourth and fifth instar *R. prolixus* nymphs were derived from the same colony and strain, original from a collection in Casanare department in 2010 prior to any insecticide application campaigns. Fourth and fifth instar *T. dimidiata* nymphs collected in a 2017 sampling event in Socotá municipality, Boyacá department, contributed to the established *T. dimidiata* colony. Both colonies have been kept in the BCEI laboratory without insecticide resistance pressure. All the insects used for the study had the same feeding time and were from the same hatch.

### 2.3. Insecticide Paint

Inesfly 5 A IGR paint (Inesfly Corporation, Valencia, Spain) is a water-based polymer coating with slow release microencapsulated insecticide. This commercially available paint is composed of a liposoluble micro-capsulated formulation containing organophosphate-based ingredients and chitin synthesis inhibitors: Diazinon 1.5%, Chlorpyrifos 1.5%, and Pyriproxyfen 0.063%. This domestic-friendly paint is widely available in Colombia and is used for insect control inside and outside the households.

### 2.4. Laboratory Bioassay

Bioassay prototype chambers were designed and built consisting of a brick base (10 cm × 20 cm × 20 cm) and four cardboard plates (20 cm × 40 cm) as lateral barriers and sealed with tape and silicone to avoid cracks where insects could escape (Figure 2). The internal walls were covered with Vaseline® (Unilever PLC, London, United Kingdom) to avoid the triatomines resting in the walls and to ensure contact with painted brick. A total of eight bricks were painted: six with Inesfly 5A IGR diluted in water, following manufacturer’s instructions, and two controls painted with a commercial paint without insecticide. The replicates made a total of eight sampling units to evaluate survival of 40 fourth and fifth instar *R. prolixus* nymphs and 40 fourth and fifth instar *T. dimidiata* nymphs. To evaluate residual effect of the paints, the insects were exposed for 60 min by placing them inside the chambers, and posteriorly removed. Mortality was evaluated at 24 h and 48 h post-exposure. The chambers were kept and used posteriorly at three different times repeating the same process to evaluate the long-term effect: T1 was the initial application, and insects were placed in each chamber 24 h after paint applications. T2 insects were placed at 3 months after paint applications, and T3 at 6 months. Evaluation of mortality was made through mechanical contact to assess stimulus response. For each repeat, 80 nymphs were used to evaluate the effectiveness of the paint using 10 insects of the same species in each chamber.

### 2.5. Field Application

Between November and December 2019, a mixed methods intervention study was performed to evaluate the effectivity of the paint to reduce the presence of triatomine insects. Inesfly 5 A IGR water-based paint product was applied to 24 selected households’ inside and outside walls. Paint was diluted 50% using distilled water for porous surfaces for a first layer and diluted 90% for a second application five hours later. Paint was applied using hand paint-rollers with ensured uniformity on the entirety of the surfaces. The households were monitored 12 months after paint application and homeowners were given a questionnaire. The questionnaire evaluated different risk factors such as house construction material, peridomicile characteristics and animal presence in the household, the presence of triatomines in the year following the insecticidal paint application, whether the individuals living in the household had experienced any adverse reactions, and demographic information. Field technicians performed a visual evaluation of the presence of triatomines revising the houses using the man/hour method, examining households for evidence of triatomine presence and triatomine eggs to determine household infestation, and/or colonization [36].

### 2.6. Ethics Statement

The study was conducted according to the guidelines of the Declaration of Helsinki and approved by the Ethics Committee for Animal Experimentation at the Universidad de Antioquia. Acta 112 de 14 de Agosto 2017.

## 3. Results

### 3.1. Laboratory Bioassay

As noted in Table 1-a, *R. prolixus* nymphs were susceptible for T1 at 24 h and 48 h; T2 at 24 h and 48 h; and T3 at 24 h and 48 h, with 100% mortality in all three experimental treatment groups. No mortality (0%) was ever noted in any control group. Treatment susceptibility was noted for the treated groups. The *T. dimidiata* group, however, showed slower mortality rate, as noted in Table 1-b. *T. dimidiata* nymphs demonstrated 80% susceptibility in one T1 replicate at 24 h (93.33% average for T1–24 h) and 90% susceptibility in one T2 replicate at 24 h (96.67% average for T2–24 h). All other experimental groups demonstrated susceptibility with 100% mortality.

### 3.2. Field Application

Results one year post application were collected through individual questionnaires to household members: responses were obtained for 21 of the 24 selected households. As noted in Table 2, household construction material was varied, with some being well-constructed (approximately half had brick and corrugated roof panels) and the other half had natural environmental housing material (e.g., mud, sticks, etc.). Domestic and agricultural animals were common in the peridomestic environment (91% of homes reported at least one animal). Almost all homes had ecological risk factors supportive of triatomine presence in the peridomestic environment (e.g., cracks in house walls, chicken coop, wood piles, trapiche mills, etc.). Lastly, sylvatic *Trypanosoma cruzi*-competent animal species were reported in most household surrounding areas: 95% of households reported opossum sightings; 62% reported rodents; and 24% reported bats. In summary, these homes represented epidemiologically high-risk scenarios for triatomine establishment.

Overall, 14 (66.7%) households responded having ever seen a triatomine, and the same proportion responded having seen a triatomine in the past year. From those that had seen a triatomine in the past year, 100% of them had seen them inside the household, in the peridomestic environment, and/or in the immediate sylvatic environment, without colonization. All respondents reported a reduction of triatomines in the year following insecticide application (100%), particularly around the household perimeter (95.2%). Triatomines inside the home were seen in: the entrance (85.7%), the bedroom (57.1%), the hallway (50.0%), the living room (14.3%), the kitchen (7.1%), the bathroom (7.1%) and the dining room (7.1%). In the peridomicile, triatomines originated from the chicken coop (92.9%), piles of rocks and/or wood (92.9%), stubble (71.4%), bird nests (35.7%), cactus (35.7%), and piles of bricks (21.4%). From the sylvatic environment, triatomines originated from: a crop field (57.1%) and nearby forest (42.9%). Lastly, allergic reactions were reported by 14.3% (3/21) of the respondents following insecticide paint application of their homes. Of these three homes with reported allergic reactions, two noted eye irritation and two noted dizziness after paint application. Additionally, field technicians did not observe any insects nor triatomine eggs presence that would bring evidence of household colonization.

## 4. Discussion

This is the first study evaluating insecticide paint effect in a laboratory and natural condition setting in Colombia. Results from this study indicated that insecticide paint Inesfly 5A IGR has a lasting effect on the mortality of *R. prolixus* and *T. dimidiata*, under laboratory conditions. These results are comparable to other evaluations in *Triatoma infestans,* strengthening the evidence of this product as a control measure [38,39]. *T. dimidiata* showed a reduced susceptibility to insecticide paint when measured at 24 h after contact, however, mortality was 100% at 48 h in all the evaluations, suggesting a great performance of the insecticidal paint. Indoor Residual Spraying (IRS) has been effective against *R. prolixus,* and *T. dimidiata* is now spreading in areas certified free of *R. prolixus,* which highlights the importance of control measures effective for *T. dimidiata,* an emerging species [15,40].

The reemergence of vectors in locations where the primary species was previously deemed eradicated raises concerns [15,16]. The current study undertook complementary controlled (laboratory-based) and real-world (field-based) investigations to ascertain the susceptibility of two prominent Colombian triatomine species to an innovative, commercially available insecticide. Our laboratory investigation revealed high susceptibility to Inesfly’s water-based insecticide indoor/outdoor paint. *R. prolixus* demonstrated high susceptibility at both 24 and 48 h, and *T. dimidiata* exhibited overall great susceptibility, with 100% mortality at 48 h. These findings were further substantiated in field trials. Field technicians did not observe any triatomines when attending houses, although homeowners reported triatomine sighting, and overall reduction was reported when comparing those results to previous investigations [21]. In summary, this proof-of-concept study confirms a positive impact of microencapsulated insecticidal paint among natural triatomine species that warrants continued surveillance to mitigate Chagas disease human transmission.

Our study is not the first one to investigate the application of insecticidal paints as an optional alternative to insecticide spray, with limited studies having displayed insecticide paint potential across the South American continent. Amelotti et al. evaluated insecticidal paint efficacy over a one-year period on *Triatoma infestans* populations in the Gran Chaco region of South America, and identified 81–100% susceptibility to painted surfaces using pyrethroid and organophosphate formulations [38]. Insecticidal paint evaluations in households of the Bolivian Chaco, using the same commercial paint, showed a demonstrable effect on mortality of *T. infestans* 34 months after application [34], and maintained infestation below 2% after 32 months [39]. The residual effect of the insecticidal paint has been reported to be longer than the residual effect of indoor spraying, suggesting this as an economic alternative for vector control. In the literature results on experimental settings that vary depending on the formulation of the paints, organophosphate formulated paints show greater mortality than pyrethroid formulation, supported by studies that show a better performance of organophosphate formulation over other insecticides [41]. The study from Maloney et al. found that the use of insecticidal paint composed by diazinon (1.5%), chlorpyrifos (1.5%), and pyriproxyfen (0.063%) produced significantly greater mortality than exposure to deltamethrin [31]. There is, however, an important controversy: the use of chlorpyrifos has been deemed successful in reducing pests and producing high insect mortality, however scrutinous investigations found associations with its use to neurological effects, endocrine disruption, and cardiovascular disease, which led to the ban of chlorpyrifos use in the United States and other countries [42]. Nevertheless, the method of delivery of chlorpyrifos through microencapsulated particles is designed to have significantly less toxicity in large vertebrates, and thus avoids harmful health outcomes [43]. Despite these published investigations, the overall literature is limited, and our current study fills this knowledge gap by adding pertinent contemporary results on the two leading vectors of public health importance in Colombia. Future investigations should consider evaluating further the safety of these microencapsulated paints on human health.

*Triatoma dimidiata* is an emerging secondary vector of public health importance. Like several other countries, the Boyacá department in Colombia has noted an increasing trend of this species in the peridomestic environment following PAHO’s 1990–2000’s *R. prolixus* elimination campaign. Historically, *T. dimidiata* has demonstrated a greater ability to shift between sylvatic and domestic environments compared to *R. prolixus,* and thus has an evidenced survival advantage. Moreover, a wide genetic variability could be beneficial for triatomines in areas such as Boyacá with diverse microclimates incorporating the insect’s most favorable conditions [44]. This versatility has allowed *T. dimidiata* to intrude and colonize households that eliminated *R. prolixus* modifying *T. cruzi* transmission risk distributions [21]. Insecticide spraying along with physical and chemical barriers has proved to reduce house infestation to a greater extent than spraying alone, suggesting increased efforts are needed to effectively control for *T. dimidiata* [45].

Today’s insecticidal paint manufacturers are able to create non-toxic continual-release formulas that allow for sustainable vector control in resource limited settings [43]. The average indoor residual spray lasts 6 months–in sharp contrast to our results, which demonstrate efficacy of the interventional insecticide paint up to 1-year post-application in a field study. Application of insecticidal paints on infested household walls has shown to be beneficial for the owners and evidence suggests strong community uptake [46]. Similar to a prior Honduran study [46], our results suggest great adherence, with a reported reduction of the presence of triatomines inside and along the household perimeter. A recent Peruvian community-based intervention using Datta and Mullainathan’s ‘behavioral design’ theory demonstrated that household-based vector control interventions, which empower the home owner, have long-last benefit for Chagas disease reduction [47]. Bolivian and Guatemalan eco-health interventions that educate and involve the community also found higher efficacy of national insecticide residual spraying programs [48,49]. Therefore, the current study’s high community participation is a notable finding in itself. Further, the fact that this insecticide can be used to treat other local agricultural pests (e.g., palm weevil) enhances the probability that this commercially available, low-cost solution will be used in between government vector control campaigns. Lastly, the novel combination of chemicals used for insecticidal paint can be a solution to single chemical resistance [24].

The results of this study showed the elimination of infestation and only sporadic appearances of insects around the households, as reported by homeowners. However, although insecticidal paint could be an alternative to insecticide spraying, other factors are involved in the effectiveness of maintaining triatomines averted from homes. Some conditions surrounding the households favor triatomine habitat, peridomestic animals keep triatomines fed around the houses, and piles of wood, rocks and other debris provide a place to hide for the insects [50]. Thus, vector control interventions should not be limited to only one type of intervention but allow for the use of effective traditional methods and novel alternatives to obtain more effective results. Additionally, vector control interventions should rely on complementary educational programs that allow for reduction of triatomine shelter in the household surroundings to decrease the risk of Chagas disease transmission.

A few investigation limitations are worth noting. The selection of homes for interventional paint application was restricted to households that had suitable conditions for paint treatment. Some of the households in the region are not adequate for water-based paints, nor have a surface suited for paint application, therefore the study findings and utility are limited to construction of wood, brick, and adobe. Moreover, a control group was unavailable, as for ethical reasons no households were left untreated, thus homes that did not receive the insecticide paint received indoor residual spray. These homes were not evaluated by survey due to logistical feasibility. However, the results from this proof-of-concept study suggest that paint application would reduce the triatomine presence and prevent triatomine dwelling inside the households. Additionally, our study evaluated for the first time the residual effect of the insecticidal paint on the two major species in the department of Boyacá, *R. prolixus* and *T. dimidiata,* increasing the external validity of the laboratory results. Lastly, additional species present in Boyacá department were not included which warrant future investigation: *T. venosa, P. geniculatus,* and *P. rufotuberculatus.*

## 5. Conclusions

In conclusion, this preliminary study demonstrates the efficacy and utility for insecticidal paint. Susceptibility to insecticidal was high, however insecticidal paint should be used in combination with other vector control techniques for greater efficacy. While insecticide spraying requires repeating the process every six months, insecticidal paint requires less frequency of application, with some reports showing more than 2 years of residual effect. Additionally, the low cost of this intervention would allow for greater uptake in the targeted population. This promising solution could maintain a reduced incidence of Chagas disease and continue the battle to eliminate vectorial transmission in Colombia.

## Figures and Tables

**Figure 1 insects-13-01061-f001:**
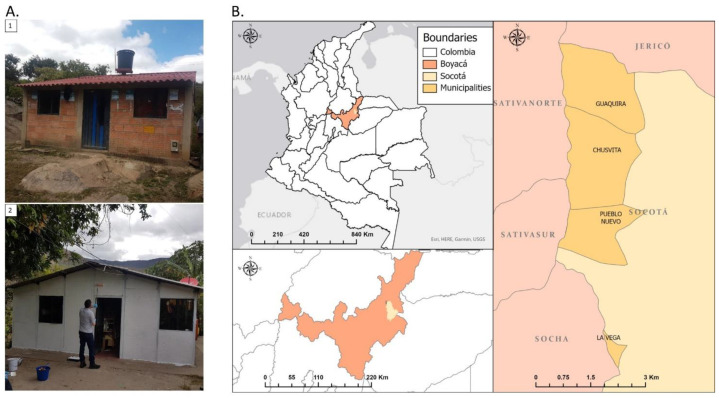
(**A**). Examples of selected households based on suitability before applying the paint (1) and after applying the paint (2). (**B**). Map of the selected municipalities from the Socatá region, Boyacá department, Colombia.

**Figure 2 insects-13-01061-f002:**
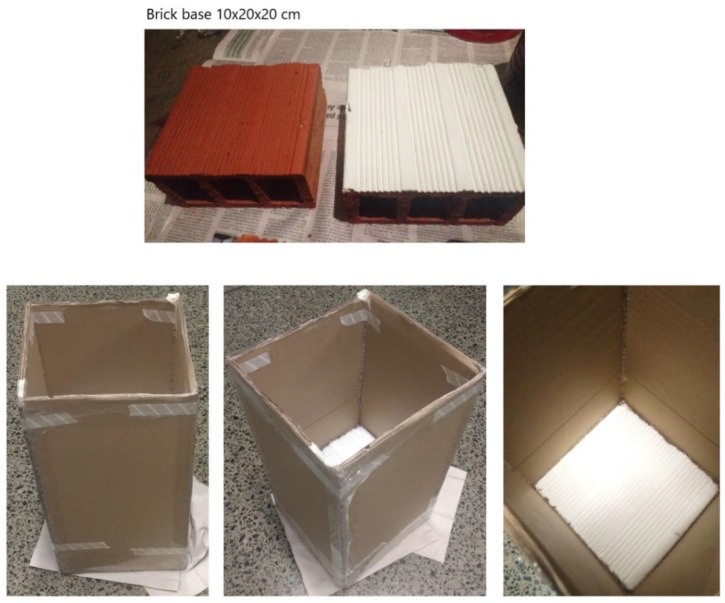
Bioassay prototype chambers for triatomine exposure to micro-encapsulated insecticidal paint. Triatomines were exposed for 1 h inside the chambers for follow-up at 24–48 h post exposure.

**Table 1 insects-13-01061-t001:** Mortality% to treated and control chamber prototypes in *R. prolixus* and *T. dimidiata* for each application time.

a. *Rhodnius prolixus*							
		T1		T2		T3	
	n	24 h ^1^	48 h	24 h ^1^	48 h	24 h ^1^	48 h
Control	10	0	0	0	0	0	0
Replica 1	10	100	100	100	100	100	100
Replica 2	10	100	100	100	100	100	100
Replica 3	10	100	100	100	100	100	100
**b. *Triatoma dimidiata***							
		**T1**		**T2**		**T3**	
	**n**	**24 h ^1^**	**48 h**	**24 h ^1^**	**48 h**	**24 h ^1^**	**48 h**
Control	10	0	0	0	0	0	0
Replica 1	10	80	100	90	100	100	100
Replica 2	10	100	100	100	100	100	100
Replica 3	10	100	100	100	100	100	100

^1^ The first evaluation showed some insects had died and some were in a “knock-out” state–not moving, or having difficulty walking or moving antennae–after a stimulus response. All knock out insects were dead in the 48 h evaluation.

**Table 2 insects-13-01061-t002:** Description of household characteristics.

	Description	N * (%)
Wall materials	Adobe	11(52.4)
	Brick	10 (47.6)
	Wood	2 (9.5)
	Mud and Bahareque	1 (4.8)
Roof materials	Zinc	10 (47.6)
	Mud	16 (76.2)
	Asbestos cement	6 (28.6)
Floor materials	Ground/soil	10 (47.6)
	Cement	19 (90.5)
	Tiles	4 (19.1)
	Wood	1 (4.8)
Features surrounding the household	Chicken Coop	19 (90.5)
	Barn	3 (14.3)
	Pigsty	6 (28.6)
	Oven	10 (47.6)
	Mill	6 (28.6)
	Wood pile	20 (95.2)
	Rock pile	19 (90.5)
Other	Any sight of wall cracks	15 (71.4)
	Average number of bulbs inside the home [Mean ± SD [37]]	3.76 ± 1.48
	Average number of bulbs outside the home [Mean ± SD]	1.14 ± 0.91
Domestic animal presence	Homes with animals	20 (95.2)
	Dogs	19 (90.5)
	Chickens	19 (90.5)
	Pigs	8 (38.1)
	Cows	15 (71.4)
	Horses	9 (42.9)
	Cats	17 (81.0)
	Domestic birds	8 (38.1)
	Domestic rabbits	1 (4.8)
Have seen wild animal in the surroundings	Opossum	20 (95.2)
	Rodent	13 (61.9)
	Bat	5 (23.8)
	Pigeon	8 (38.1)
	Rabbit	2 (9.5)

* Percentages may add up to more than 100% due to respondent’s multiple answer option.

## Data Availability

Data presented in this study are available upon request from the corresponding author.

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
