# Peer review of "Laboratory Evaluation and Field Feasibility of Micro-Encapsulated Insecticide Effect on Rhodnius prolixus and Triatoma dimidiata Mortality in Rural Households in Boyacá, Colombia"

_insects, 2022, doi:10.3390/insects13111061_

Round 1

Reviewer 1 Report (Previous Reviewer 1)

The authors have performed a good job in revising the manuscript.

Author Response

Thank you so much for the review comments.

Reviewer 2 Report (Previous Reviewer 2)

My questions have been resolved. There was is significant improvement in the article. So I'm favor of publishing the article. 

Author Response

Thank you so much for the reviewer's comment.

Reviewer 3 Report (New Reviewer)

The authors have made a potentially useful study, but personally the concerns I have are over the potential long-term exposure of people to the three pesticides encapsulated in the paint. I suspect they are mostly unknown but could be discussed further in the discussion.

There were a few word mistakes but otherwise the writing was clear, and the data satisfactory.

Author Response

Thank you so much for the reviewer's comment.

We followed the suggestion and added an additional sentence in lines 291-293 addressing future directions evaluating the safety of micro-encapsulated paints.

This manuscript is a resubmission of an earlier submission. The following is a list of the peer review reports and author responses from that submission.

Round 1

Reviewer 1 Report

General comments

    The work of Gual-Gonzalez and coworkers deals with an interesting subject: the effect of micro-encapsulated insecticide in a paint against Chagas disease vector species of epidemiological significance. For this purpose, the authors performed a laboratory assay and a field survey. The objectives are clear and straightforward and the findings are relevant to the addition of novel approaches to fight Chagas disease spread. Nevertheless, it is not clear why the authors chose to perform a questionnaire to the house owners and not a proper entomological evaluation, actively searching the insects before and after the application as done in DOI: 10.1186/s13071-015-0762-0. This fact renders the results of the field part of the study largely subjective. This point must be justified and explained.

   The authors also need to tone down some expressions and conclusions regarding the ability of the paint to prevent colonization and the fact that they are studying resistance.

Specific comments

Simple summary:

- Page 1, lines 21-22: Due to the way the study was conducted, the findings did not conclusively “demonstrate the ability of this paint to prevent colonization”. Please tone down the expression.

- Page 1, line 22: Consider replacing "Findings demonstrate the ability of this paint to prevent colonization and an showed…" by " Findings demonstrate the ability of this paint to prevent colonization and showed…".

Abstract:

- Page 1, lines 31-32: Consider replacing "…in laboratory conditions and its applicability rural communities…" by "…in laboratory conditions and its applicability in rural communities…".

-Page 1, line 41: Replace “pain” by “paint”.

Introduction

- Page 2, lines 60-63: The way the sentence is written appears that the Indoor Residual Spraying has been used to interrupt transfusion-based transmissions. Please rewrite.

- Page 3, line 95: Replace “herbicides” by “pesticides”.

Discussion

- I believe that a more comprehensive description of the paint employed would be useful for the non-specialized audience.

- Page 7, lines 230-234: Some relevant literature is missing, in particular the work of Gorla et al (2015, DOI: 10.1186/s13071-015-0762-0) since they use a similar, although more comprehensive approach on vectors of Chagas disease.

- Page 8, lines 246-248: the authors claim that “…T. dimidiata exhibited incipient resistance at 24h under controlled laboratory conditions, but overall, 100% mortality at 48h.”. Nevertheless, the very definition of resistance implies insect survival, not delay in dying (Zhu KY. Insecticide resistance. In: Capinera JL, editor. Encyclopedia of entomology. 2nd ed. New York: Springer; 2008. p. 1979–81 and https://irac-online.org/training-centre/resistance/). This expression and that of line 254 must be revised.

- Page 9, lines 303-304: The authors claim that “Results of this study showed elimination of infestation and only sporadic appearances of insects around the households”. Nevertheless, given the limitations of the field study, that claim should be toned down.

Conclusions

- Page 9, lines 330-331: Consider replacing “…chemical resistance was noted therefor…” by “…chemical resistance was noted, therefore”.

Reviewer 2 Report

I would like to congratulate you for the article that deals with the evaluation of the effect of insecticide paint on two species of triatomines under laboratory and field conditions. Below I make a few observations that I think are relevant. 

As for the material and methods I alk:

I line 325, the authors discuss the increase in external validation of their results. Ask: Wouldn't external validation depend on the draw of the properties selected in conditions to receive the insecticide paint ?

In the results, table 1, the authors describe the answers given by the selected households to answer the questionnaire. The physical characteristics of households and the presence of peridomestic or even wild animals in the vicinity represent a spatial and trophic niche for triatomines, which would provide shelter respectiveley for triatomines to develop their life cycle. Regarding table 1 I ask: In what way could the variable average number of light bulbs found inside and outside the house be discussed in relation to the presence of triatomines. We know that insects, hematophagous or not, are attracted to light. How do you discuss this variable? What is the importance of this variable in this questionnaire?

I present some more suggestions for discussion of the article. These are doubts so they sometimes present themselves in the form of questioning. Maybe they are also limitations in this article. How the selected aggregates also answered about the presence or absence of triatomines after apllications of insecticide paint. I wonder if these respondentes were able to differentiate between a Hemiptera of the Reduviidae Family, hematophagous or predatory entomophagous or Hemiptera phytophagous, Coreidae, Pentatomidae or others families. Were the triatomines found by the respondentes after the application of the insecticide paint confirmed or not by entomologists? I alk because there are morphological similarities between these and for a layman there would be difficulty in differentiating unless he was trained. It would be interesting if entomological research had tajen place after the application of insecticide paint. As the selected households are in the rural area could occur the invasion of insects of the order Hemiptera with phytophagous habitats and confuse the respondentes. 

These are the observations and questions I make for the article.